# Advances in 3D Culture Models to Study Exosomes in Triple-Negative Breast Cancer

**DOI:** 10.3390/cancers16050883

**Published:** 2024-02-22

**Authors:** Neelum Aziz Yousafzai, Lamyae El Khalki, Wei Wang, Justin Szpendyk, Khalid Sossey-Alaoui

**Affiliations:** 1MetroHealth System, Cleveland, OH 44109, USA; nay10@case.edu (N.A.Y.); lamyae.elkhalki@case.edu (L.E.K.); wxw363@case.edu (W.W.);; 2Department of Medicine, Case Western Reserve University, Cleveland, OH 44106-4909, USA; 3Case Comprehensive Cancer Center, Cleveland, OH 44106-7285, USA

**Keywords:** exosomes, 3D culture, breast cancer, TNBC, organoids, extracellular vesicles, exosomes, EMT, immuno-oncology, tumor microenvironment, metastasis

## Abstract

**Simple Summary:**

Breast cancer comes in different types, making it hard to treat effectively. One particularly aggressive type, called triple-negative breast cancer, is tough to target with current treatments. Scientists use advanced methods like 3D cultures, which mimic human tissue better than traditional lab methods, to study breast cancer. These 3D cultures help understand how tiny communication structures called exosomes affect cancer growth, spread, and response to therapy. Exosomes are like messengers between cells and can influence cancer’s behavior and response to therapy. In this the review, we seek to discuss the effect of 3D cultures on the function of exosomes, which we believe could play an important role in improving exosome-mediated cancer treatment by delivering drugs more precisely and studying how cancer becomes resistant to treatment. This research could lead to better therapies and help us understand how cancer spreads and resists treatment.

**Abstract:**

Breast cancer, a leading cause of cancer-related deaths globally, exhibits distinct subtypes with varying pathological, genetic, and clinical characteristics. Despite advancements in breast cancer treatments, its histological and molecular heterogeneity pose a significant clinical challenge. Triple-negative breast cancer (TNBC), a highly aggressive subtype lacking targeted therapeutics, adds to the complexity of breast cancer treatment. Recent years have witnessed the development of advanced 3D culture technologies, such as organoids and spheroids, providing more representative models of healthy human tissue and various malignancies. These structures, resembling organs in structure and function, are generated from stem cells or organ-specific progenitor cells via self-organizing processes. Notably, 3D culture systems bridge the gap between 2D cultures and in vivo studies, offering a more accurate representation of in vivo tumors’ characteristics. Exosomes, small nano-sized molecules secreted by breast cancer and stromal/cancer-associated fibroblast cells, have garnered significant attention. They play a crucial role in cell-to-cell communication, influencing tumor progression, invasion, and metastasis. The 3D culture environment enhances exosome efficiency compared to traditional 2D cultures, impacting the transfer of specific cargoes and therapeutic effects. Furthermore, 3D exosomes have shown promise in improving therapeutic outcomes, acting as potential vehicles for cancer treatment administration. Studies have demonstrated their role in pro-angiogenesis and their innate therapeutic potential in mimicking cellular therapies without side effects. The 3D exosome model holds potential for addressing challenges associated with drug resistance, offering insights into the mechanisms underlying multidrug resistance and serving as a platform for drug screening. This review seeks to emphasize the crucial role of 3D culture systems in studying breast cancer, especially in understanding the involvement of exosomes in cancer pathology.

## 1. Introduction

Breast cancer, the foremost cause of cancer-related deaths globally, exhibits distinct subtypes with variations in pathology, genetics, and clinical features, predominantly affecting women [1]. Despite advancements in breast cancer treatment methods, its histological and functional heterogeneity poses a significant clinical challenge. Current evidence indicates that these subtypes manifest diverse genetic alterations, where cell adhesion and secreted extracellular proteins play crucial roles in shaping the tumor microenvironment [2,3]. The identification of targeted therapies for breast cancer based on genomic alterations remains challenging due to extensive molecular heterogeneity, particularly in metastatic cases [4]. Triple-negative breast cancer (TNBC), characterized by aggressiveness and metastatic potential, presents additional challenges due to the lack of targeted therapeutics [5].

Maintaining cancer heterogeneity in culture is vital for understanding genotype–phenotype relationships, influencing the success of targeted agents. In recent years, advanced 3D culture technologies have provided more representative models of both healthy human tissue and various malignancies [6,7]. Organoids and spheroids, intricate three-dimensional structures resembling organs, have been instrumental in advancing cancer research [8]. Sato and his team pioneered the development of 3D epithelial organoids, setting the stage for organoid development from various human epithelial tissues, including the colon, prostate, stomach, liver, lungs, pancreas, esophagus, endometrium, taste buds, salivary glands, fallopian tubes, and breast tissue [9].

In cancer studies, 3D cultures serve as intermediate models between 2D cultures and in vivo studies, offering a more accurate representation of in vivo tumors’ characteristics. This includes aspects such as cell-to-cell and cell-to-extracellular matrix contacts, cellular layering, hypoxia, and gradients of nutrients and pH [10,11]. Evaluating extracellular vesicles (EVs) in vitro through 3D culture systems provides distinct advantages over in vivo assessments. Despite the challenges associated with monitoring EVs in vivo, a considerable amount of research has focused on this approach [12,13,14].

Poor prognosis in breast cancer is linked to rapid tumor progression, interactions within the tumor microenvironment, and the dissemination of tumor cells through blood circulation to form metastatic tumors in other organs [15]. Studies have shown that tumor-derived extracellular vesicles, specifically exosomes (50–150 nm diameter), released from cells can transfer bioactive molecules to neighboring or distant cells via body fluids such as the bloodstream [16]. Due to their unique trafficking characteristics, exosomes are emerging as potential therapeutic RNA delivery vehicles for early diagnosis, prognosis, and targeted therapy development [17,18,19]. The process of harvesting exosomes is influenced by the culture system, isolation method, and purification process employed [20,21]. Accumulating evidence suggests that a three-dimensional (3D) culture system yields more exosomes compared to the traditional two-dimensional (2D) system [22]. Moreover, exosomes produced by 3D culture exhibit enhanced therapeutic effects through the transfer of specific cargoes, emphasizing their potential in clinical cancer studies [19]. This review aims to underscore the pivotal role of these studies in comprehending the involvement of exosomes in TNBC tumor physiology and identifying potential applications in clinical cancer research.

## 2. Exosomes in 2D versus 3D Culture System

In recent decades, an expanding body of cancer studies has brought to light that exosomes, small nano-sized molecules ranging from 30 to 1020 nm, represent the primary class of extracellular vesicles secreted by breast cancer and stromal/cancer-associated fibroblast cells into the extracellular milieu and tumor microenvironment [23]. Under stressful conditions, exosomes released by breast cancer cells have been linked to an increase in the invasive and metastatic potential of these cells [24]. The unique biogenesis of exosomes involves distinct intracellular regulatory processes that equip them with specific cargos and, consequently, diverse biological functions. Fibroblast-derived exosomes, for example, have been demonstrated to enhance breast cancer metastasis and motility via the Wnt pathway [25]. Mesenchymal stem cell (MSC) exosomes have displayed remarkable efficacy in cancer metastasis, tissue repair, and regeneration across various organs, including the liver, lungs, cartilage, myocardium, brain, spinal cord, kidney, and breast [26,27,28,29,30].

In investigating the tumor microenvironment and cell-to-cell communications, the conventional in vitro method is the 2D culture, a “gold standard” approach for cultivating monolayer cells and isolating exosomes. This choice is primarily supported by its simplicity, reproducibility, and cost effectiveness. However, several studies have indicated that the 2D model, in terms of mechanical and biochemical features, falls short of accurately mimicking the pathophysiology and features of three-dimensional (3D) confirmation with specific architecture found in in vivo solid tumors [10].

Conventional 2D cell culture methods particularly struggle to replicate the density and heterogeneity of clinical tumors as observed in vivo. Consequently, over recent years, alternative 3D culture models have been under development. The shift to 3D culture environments aims to address the disparities between monolayer cells under 2D conditions and the in vivo conditions where cells grow in three dimensions, impacting cell morphology, cell signaling crosstalk, interactions with the extracellular matrix (ECM), drug screening, and the understanding of tumor progression, invasion, and metastasis [14,31,32,33,34].

Lee and colleagues established the most suitable 3D cell culture methods for studying stem cells and the microenvironment of breast cancer, developing a 3D culture system for the mammary epithelium that enables the generation and long-term expansion of three-dimensional epithelial organoids [35]. When compared to 2D cultures, 3D culture environments revealed altered dynamics and molecular contents of breast cancer exosome secretion, including RNA [36]. The findings from the study suggest that exosomal RNA content derived from 3D culture could replicate in vivo tissue-derived exosomal RNAs. Studies have identified microRNAs that are more prevalent in 3D-culture-derived cervical cancer exosomes compared to those derived from 2D culture [14]. In the 3D cell culture system, interactions between cells and the extracellular environment closely resemble those in tumor tissue masses, as depicted in Figure 1. These interactions play a pivotal role in cell differentiation, viability, protein expression, response to stimuli, drug metabolism, drug resistance, and other cellular functions [37,38].

Kim and colleagues reported that the efficiency of exosome secretion could be increased using 3D spheroid culture of MSCs from human bone marrow, further involving key parameters affecting exosome secretion, including cell spheroid size, cell morphology, and cell density [39]. While other studies have suggested that exosome secretion and the transfer of small molecular cargoes (RNAs and proteins) are influenced by various factors, including parent cell morphology, physiological and pathological conditions, and stimuli from microenvironments [40,41]. Exosomal miRNAs are utilized as biomarkers for cancer diagnosis. Moreover, tumor-derived exosomes containing miRNA, such as the miR17-92 cluster, were found to induce endothelial migration and tube formation [42], while fibroblast-derived exosomes were found to contribute to the progression of breast cancer metastasis and motility through the Wnt pathway [30], a pathway that is established to play a significant role in stem cell maintenance, tissue patterning, and tumorigenesis [25]. Another study has suggested that the efficiency of exosome secretion can be enhanced by preconditioning cells using 3D culture methods, involving genetic manipulation, exposure to hypoxia, increased intracellular calcium [43,44,45], and treatment with specific growth factors. These preconditioning procedures can increase the yield of exosomes, thereby improving their clinical applications [46].

## 3. Exosomes and 3D Culture Organoid Model

One significant stride in cancer stem cell research over the past decade has been the emergence of “organs” known as organoids. In the realm of classical developmental cancer biology studies, “organoid” refers to a self-organizing 3D structure derived from stem cells [47]. Organoids, which faithfully mimic the in vivo architecture and cell differentiation of the original tissue in mammals, utilize defined developmental cues. The phenotypes of stem cells can be differentiated using specific conditional media [48]. Three-dimensional organoid culture systems are extensively employed as a disease model in studies, encompassing the expansion of tumor-initiating cells [49], investigation into invasion, progression, and metastatic developments, as well as drug screening to understand cell responses to irradiation [50]. In vitro, tumor cells exhibit the capability to form spheroids, colonies, tumoroids, and organoids, all of which manifest increased cancer stem cell properties [51]. Additionally, to comprehend the universal transcriptional mechanisms occurring under 3D culture organoids, certain differentially expressed genes are identified in cytokine and extracellular vesicle (EV) biogenesis, various types of cytoskeleton, and the extracellular matrix (ECM) like fibronectin facilitates to enhanced the 3D culture properties of breast cancer [52,53,54].

While the value of 3D cell culture in studying tumor cells is well established, Rocha and colleagues showed that tumor cells cultured in 3D conditions secrete EVs that contain protein cargo and miRNAs that are distinct from those contained in 2D EVs, emphasizing the need for further investigation into EV biogenesis markers [55]. Consequently, dynamic aggregate tumoroids alter EVs biogenesis in human MSCs by potentially stimulating endosomal sorting complexes required for transport (ESCRT) dependent and independent signaling pathways [53].

Since its identification, the spheroid model of ovarian cancer has proven to be particularly relevant for in vitro studies, providing insights into how EVs contribute to carcinogenesis through stem cell initiation and differentiation. This system offers enhanced reliability and stability, mirroring in vivo situations more closely [56]. Previous studies have identified tumorspheres and organoids as morphological markers of cancer stem cells (CSC), with markers such as CD44 being regulated by ESRP1 in breast cancer cells [57], while TGF-β has been shown to suppress the expression of ESRP2 by inducing ZEB1 and ZEB2 [58].

Furthermore, a recent study evaluating the influence of hypoxic conditions on a 3D matrix as an in vitro model for EVs, exposed to fibrin, revealed that EV concentration increased with oxygen deficiency [59]. The study also disclosed that exosome concentration from 3D cultures could be 3 to 6.7 times higher than that from 2D cell cultures, depending on the O_2_ level during the incubation period. Despite this, few studies have delved into EV secretion from 3D scaffolds or patches [60]. Collectively, these studies validate the potential use of organoids, spheroids, and other in vitro 3D systems, primarily organoids, to explore the role of exosomes in breast cancer and other cancers [61]. In this context, recent and previous studies comprehensively cover the framework for other inquiries with 3D culture systems, and the functions of exosomes, and are detailed in Table 1.

## 4. Exosomes Derived from 3D Culture Improve Therapeutics Effect

The potential impact of exosomes in the field of cancer biotherapy is extensive. Exosomes may also serve as a versatile vehicle for administering cancer treatment. Exosomes play pivotal roles in transferring proteins, nucleic acids, and lipid molecules, positioning them as a promising avenue for effective cancer therapy and targeted antigen/drug carriers. Notably, studies in cancer have revealed that exosomes released from MSCs of various sources differentially influence tumor progression, invasion, and metastasis [74]. The therapeutic efficacy of MSC-based therapies, mediated through exosome-mediated paracrine secretion of cytokines and growth factors, has been increasingly demonstrated. A major class of MSC exosomes inherently possesses therapeutic potential due to their unique cargoes, making them a promising framework for cell-based therapies. These exosomes can act as nature’s own delivery tools, serving as biotherapeutics and mimicking the functions of cellular therapies without the side effects of embolism and proliferation. The composition of 3D exosomes is summarized in Table 2.

Studies, such as the one by Litao Yan and Xing Wu, have reported that 3D exosomes exhibit greater effects than 2D exosomes in cartilage repair in a rabbit cartilage defect model. However, the therapeutic outcome fell short of renewing normal cartilage function, primarily due to the limited curative time of only 4 weeks [22]. Recognizing this limitation, the potential of 3D models for advancing new anticancer approaches has been consistently demonstrated over time [92,93]. The 3D culture of MSC exosomes emerges as a potential therapeutic strategy for pro-angiogenesis through the activation of the HMGB1/AKT signaling pathway, as reported by Gao et al. [94]. Additionally, eliminating exosomes from the circulatory system stands out as an attractive therapeutic option to modify the metastatic effects of exosomes in breast cancer and other cancers. Recent studies have explored the use of exosomes to deliver cargo such as nucleic acids, drugs, miRNAs, and antigens, presenting a promising avenue for treating tumor progression and metastasis. Exosomes also hold the potential to deliver siRNAs for regulating gene expression, targeting specific proteins with low toxicity [95,96].

However, despite extensive investigations over the past decade aimed at elucidating extracellular vesicle functions, many characteristics and mechanisms remain unclear. Some studies present contradictory results, with extracellular vesicle types, such as MSC-derived exosomes, exhibiting dual functions—both inhibiting and promoting tumor growth [97,98]. Such discrepancies may stem from differences in cell culture methods and purification protocols used for extracting extracellular vesicles. Notably, 2D cell culture conditions significantly differ from 3D culture, influencing the content of extracellular vesicles and exosome secretion. To define and clarify the biological roles and therapeutic potential of exosomes, standardized and more accurate purification protocols are urgently needed. While differential ultracentrifugation is considered the gold standard method for exosome purification, concerns arise regarding its relatively low yield and impurity of other small molecules. Alternative methods, such as filtration-based approaches like liquid chromatography-based separation, are emerging as viable alternatives for large-scale exosome production, essential for cancer therapeutics [99].

Various strategies are being developed for cancer therapy, including targeting exosomes to inhibit their impact on diseases, manipulating their intrinsic therapeutic potential, and utilizing them for drug delivery. Inhibition of cancer malignancy in vitro has been demonstrated using a clathrin-mediated endocytosis inhibitor, chlorpromazine, targeting the mechanism of exosome miRNA-21 uptake [100,101,102]. Furthermore, the outer coat proteins of tumor-derived exosomes exhibit specific glycosylation patterns associated with other cargo proteins, regulating exosome interactions with recipient cells. Modified glycosylation of exosomal proteins emerges as a potential therapeutic strategy for cancer treatment. Notably, exosomes derived from brain endothelial cells show promise as transporters of anticancer drugs in a zebrafish model for brain cancer treatment [103].

While exosomes represent a relatively new field in cancer research, substantial interest exists in their potential application as low-toxicity and targeted inhibitors in immunotherapy, as cancer biomarkers, and for targeted therapy [104]. Tumor microenvironments play a crucial role in anti-tumor immunity, offering a newly identified strategy for cancer therapeutics. Understanding the influence of exosomes on components of the tumor microenvironment (TME) is vital for advancing the clinical benefits of tumor microenvironment inhibitors. This necessitates defining the impact of exosomes on various components of the TME using appropriate cancer models, contributing to the discovery of novel therapeutic approaches for breast cancer [105,106].

## 5. Three-Dimensional Exosomes Regulate Microenvironment in Breast Cancer

Breast cancer metastasis is a highly regulated and a complex process involving local invasion, extravasation, intravasation, transport, and colonization [107,108]. These stages necessitate key regulators, encompassing genetic, biochemical, and morphological changes within an evolutionarily conserved and interconnected program known as the epithelial-mesenchymal transition (EMT) [109,110,111,112]. Cancer extracellular vesicles, including exosomes, play a crucial role in modifying normal epithelial cells, a phenomenon termed EMT, contributing to the initiation and progression of oncogenesis [61]. In 3D cultures, exosome secretion is promoted through signal exchange, often occurring within the context of spheroids or organoids. The extracellular matrix (ECM) holds significant importance in both the homeostasis of normal breast tissues and the breast cancer microenvironment, owing to its specific composition, morphology, and cellular changes [113]. The ECM, a complex mixture of multicellular regulatory proteins, exhibits specific structural and functional properties [114]. Three-dimensional cell culture enhances the physical properties of the ECM with a complex network of extracellular bodies governing cell fate through biochemical transmission and biomechanical mechanisms [115].

MSC-derived exosomes have emerged as contributors to the metastatic niche through interactions with the stromal and matrix components, regulating immune responses targeted towards tumoroids [116]. Numerous studies have explored various components within the tumor microenvironment present in 3D spheroids, such as tumor-associated endothelial cells (TECs), cancer-associated fibroblasts (CAFs), tumor neovasculature, adipocytes, and immune cells, including tumor-associated macrophages (TAM), cytokines, tumor-infiltrating lymphocytes (TILs), and myeloid-derived suppressor cells (MDSCs) [62]. Exosomes have been shown to participate in immune suppression, affecting the regulation of CD8+ T cells, MDSCs, and Tregs [117]. Investigations of the effects of exosomes on the immune system revealed inhibitory actions on T cell activation, Th1-cytokine production in glioblastoma stem cells, and a reduction in natural killer cell proliferation [118,119].

Microenvironment heterogeneity is effectively replicated in a 3D system, such as the secretion of miRNA-21-enriched exosomes by glioblastoma stem-like cells, regulating the miR-21/VEGF/VEGFR2 signaling pathway, and promoting permeability and angiogenesis in the glioma microenvironment [120,121]. CSC-derived exosomes facilitate bidirectional crosstalk between CSCs and their metastatic niche, promoting microenvironment remodeling, resulting in tumor aggressiveness and distant metastasis [122]. These studies identify pre-metastatic niche construction as an underlying mechanism for creating a favorable environment for cancer stem cell growth. Additionally, other cancer studies identified two distinct stromal components in mammary glands, epithelial cells surrounded by intralobular fibroblasts, that are essential for tumor cell invasion and metastasis [123,124].

In-depth cancer studies have identified that EMT and CSC promote exosome secretion, possessing cancer stemness ability. These phenotypes are associated with cellular phenotypic transformation [125]. Studies published by Heather et al. hypothesize that CSC exosomes cargo lncRNAs, particularly linc-ROR, induce EMT, activate the tumor microenvironment, and contribute to the distant metastatic niche in thyroid carcinoma [120]. Dongwei et al. reported that CAFs-derived exosomes play an essential role in suppressing immune cell function in breast carcinoma through the miR-92/PD-L1 pathway in in vitro studies [30]. Moreover, it was determined that trafficking in breast cancer cells promotes autocrine Wnt11 through fibroblast-mediated exosomes, unveiling an intercellular signaling pathway where fibroblast exosomes stimulate autocrine Wnt-PCP signaling to drive aggressive invasive behavior of breast cancer in in vivo models. Components of the Wnt-PCP pathway, such as Vangl, Fzd, and Dvl, are overexpressed in several types of tumors and are implicated in tumor cell progression, migration, invasion, and metastasis [126].

## 6. The Impact of Tumor Derived Exosomes on Immune Suppression and Cancer Progression

Exosomes possess the ability to suppress the immune system, as cancer cells release these vesicles containing molecules that impede the body’s capacity to recognize and combat cancerous cells. Cancer cells modify the microenvironment and regulate the effect of the functionality of immune system, usually via pathways involving cell-to-cell communication and the release of many suppressive factors [127,128]. Within these exosomes, proteins capable of inhibiting immune cell activity, particularly T cells, are found. Moreover, exosomes carry microRNAs that regulate the expression of genes involved in immune functions are summarized in Figure 2. Additionally, their influence extends to macrophages, stimulating the production of M2 macrophages characterized by immunosuppressive and anti-inflammatory properties, facilitating tumor invasion and metastasis [129]. Exosomes transfer tumor antigens to dendritic cells in an indirect pathway but more efficient way of immunostimulation, this stimulation results in the upregulation of IL-10 and CD206 expression in naïve macrophages, leading to M2 polarization. Specific microRNAs conveyed by exosomes further contribute to suppressing macrophage immune function, playing a role in cancer progression [130,131,132,133].

Furthermore, exosomes disrupt the maturation of immature myeloid cells (IMCs) into dendritic cells and monocytes, resulting in an increased population of myeloid-derived suppressor cells (MDSCs) and other immune cells, such as B cells, monocytes can crosstalk with CAFs as well that possessing potent immunosuppressive activity [79,134,135]. These exosomes also exhibit the ability to inhibit the activity of various immune cells, including T cells, B cells, and natural killer (NK) cells [136,137]. They achieve this by containing proteins such as programmed death-ligand 1 (PD-L1), which interact with receptors on immune cells, suppressing their activity. Moreover, exosomes carry microRNAs regulating the expression of genes involved in immune function [138]. Additionally, cancer cells release exosomes promoting the activity of immunosuppressive cells like regulatory T cells (Tregs) and myeloid-derived suppressor cells (MDSCs) [139]. On the basis of previous study these cells actively inhibit the activity of other immune cells, fostering a tumor-supportive environment. The interaction between cancer cells and exosomes is intricate, contributing to various facets of tumor growth and progression [118,140]. Understanding the mechanisms underlying these interactions is crucial for developing innovative diagnostic and therapeutic strategies for cancer. Exosomes can also interfere with antigen presentation, a process vital for immune cells to recognize and respond to foreign substances like tumor antigens [141]. For instance, exosomes can carry molecules that downregulate the expression of major histocompatibility complex (MHC) molecules on cancer cells, essential for antigen presentation. A comprehensive understanding of the role of exosomes in regulating the immune response to cancer is imperative for the development of novel immunotherapies capable of enhancing the immune response and improving outcomes for cancer patients [132].

## 7. Exosomes Mediated Multidrug Resistance

Maintaining the integrity of exosomes is crucial for cancer cellular activity and intercellular communication, presenting a significant breakthrough in addressing challenges related to multidrug resistance in cancer management. Breast cancer poses a substantial hurdle to effective treatment due to the complexity of genetic alterations and the high heterogeneity of the disease [142]. The mechanisms underlying the invasive, metastatic, and aggressive ability of drug resistance remain elusive [143,144]. Over the past decade, the development of 3D models for various cancer studies, with a focus on understanding cancer stemness, aggressive behavior, tumorigenesis, and chemoresistance, has significantly expanded our comprehension of these processes [144]. Recent findings highlight the functional role of drug resistant-derived exosomal EphA2 in promoting the transmission of an aggressive phenotype and metastasis between breast cancer cells and other cellular components. Increased EphA2 protein expression in drug-resistant cancer cell-derived exosomes serves as a potential prognostic marker for drug resistance-induced breast cancer progression [145]. Exosomal miRNA-100-5P induced cisplatin drug resistance in various cancer cells, altering mTOR expression levels [146]. Liver and lung cancer induced 5-FU drug resistance through miRNA-32-5P via the AKT pathway, while breast cancer exhibited Adriamycin drug resistance delivered by miRNA-222 and colon cancer demonstrated Cetuximab drug resistance mediated by the AKT/PTEN pathway [147,148,149,150]. Breast cancer-derived exosomes containing p-glycoproteins confer chemoresistance to more sensitive recipient cells via MDR proteins, including MDR3, PABP4, and endophilin-a2 [151]. ABC transporter G2 (ABCG2), a breast cancer drug resistance protein, serves as a cancer stem cell marker [152]. Exosomes, also known as EVs, play specific key roles in the development of multidrug resistance and the transfer of genetic signals that regulate tumorigenesis, intercellular communication, and the extracellular matrix, including immune cell responses [79].

Researchers have observed that cancer patients produce an increased quantity of exosomes, exceeding 4000 trillion, compared to normal cells, leading to relapse or recurrence in cancer patients [153,154]. Exosomal miRNAs, such as miR-146a-5p, miR-222-3p, miR-151a, and cirExo-TRPCS, hold promise as diagnostic biomarkers [155,156,157]. While many drug resistance studies have been conducted in 2D culture systems, the development of 3D cultured organoids for breast cancer introduces a new approach to cancer study, facilitating the identification of diagnostic biomarkers, studying the mechanisms of tumor cell interaction with the microenvironment, and enabling drug screening, discovery, and response to drug resistance [144,158]. Although few studies have been reported using 3D culture systems for drug resistance, recent examinations have assessed chemosensitivity and chemoresistance in 3D cultured tumoroids in colorectal cancer as well [159,160].

Walsh et al. focused on organoids and tumoroids for drug resistance administration, evaluating a panel of different drugs in 3D culture tumoroids. The results identified heterogeneity in organoid response to drugs, with similar drug sensitivity observed in vitro in organoids and in vivo studies [161]. Three-dimensional culture studies are promising, considering the confirmed role of exosomes in breast cancer-related drug resistance. Tumor spheroids, a useful and versatile method, enhance exosome yield, exerting a substantial impact on cell proliferation, drug resistance, angiogenesis, migration, and tube formation through the transfer of miRNAs, bioactive proteins, and lncRNA [94,162,163,164].

In summary, the role of exosomes in the development of multidrug resistance between the tumor microenvironment (TME) and cancer cells is significant. TME, a heterogeneous population of cells in breast cancer, includes cancer-associated fibroblasts, endothelial cells, and immune cells, all interlinked with cytokines and chemokines. Breast cancer exosomes act as carriers of information, facilitating intermediate communication between cancer cells and other cellular matrices, leading to the acquisition of drug resistance [165]. With an understanding of drug resistance mechanisms via exosomes, which are loaded with various miRNAs and cargo proteins holding unique features leading to chemoresistance, researchers aim to overcome this challenge in TNBC [166]. Studies have shown that the acquired HER2-resistance of breast cancer cells can be transferred by exosomes through the dysregulation and activation of hormone-independent pathways, involving exosomal cargos such as miR-567, miR-222, miR-423-5p, miR-503, miR-155, and miR-301, influencing chemoresistance [167,168,169]. Ongoing clinical trials are exploring the use of PDL-1 in monotherapy or in combination with new targeted antitumor agents, chemotherapy, radiotherapy, and hormonal therapy to overcome drug resistance [166]. Hormonal HER-based therapy has proven highly effective in treating breast cancer, significantly improving the long-term disease-free survival of breast cancer patients.

## 8. Conclusions and Future Directions

In summary, the recent strides in 3D cancer cell culture have transformed disease modeling, showcasing its vast potential in translational medicine, biomedical research, drug screening, drug metabolism, and personalized therapy. This review underscores the pivotal role of the 3D cultured model in achieving substantial exosome yields, contributing to our understanding of stroma cells, and monitoring the morphological and pathophysiological aspects of tumor cells in cancer studies, while also serving as an efficient method for drug delivery. Exosomes, acting as nanovesicles mediating intercellular communication and modulating immune functions, stand out with their distinctive features, carrying specific protein and genetic cargo. Although 3D exosomes are a relatively new research area compared to their 2D counterparts, scientists are beginning to uncover their widespread role in cancer therapy, considering their potential use as biomarkers or in a protective and more efficient drug delivery system.

Despite these advancements, several technical challenges persist, such as the time-consuming and potentially expensive isolation and purification of exosomes from 3D cultured systems. This often necessitates further downstream purification to separate exosomes from other extracellular vesicles. Extracellular vesicles embody diverse cellular components that not only mirror the characteristics of parent cells but also influence the tumor microenvironment and cancer progression.

However, in vivo tumors exhibit extensive genetic heterogeneity, and it remains uncertain whether tumoroids/organoids can fully replicate the diverse populations originating from in vivo tumors. The establishment of 3D models from individual patient-origin tissues or cells poses both a major challenge and advantage in organoid-based culture. While 3D technology, especially in the context of breast cancer, is advancing rapidly, therapeutic mechanisms in breast cancer are still largely undefined, and the scale-up secretion of exosomes poses a significant task for clinical applications. Gaining a better understanding of true cellular crosstalk and communication in cancer biological systems requires mimicking in vivo conditions with 3D models and real-time exosome yields to better reflect cell-to-cell communication. However, obtaining in vivo tissues poses challenges due to limited access, ethical considerations, and regulatory guidelines. Thus, establishing ex vivo 3D cultured cellular models is crucial. The role of 3D exosomes in breast cancer warrants further exploration, extending to other cancer types, disease states, pathophysiological conditions, and examining the impact of culture media, including growth factors and biomaterials.

## Figures and Tables

**Figure 1 cancers-16-00883-f001:**
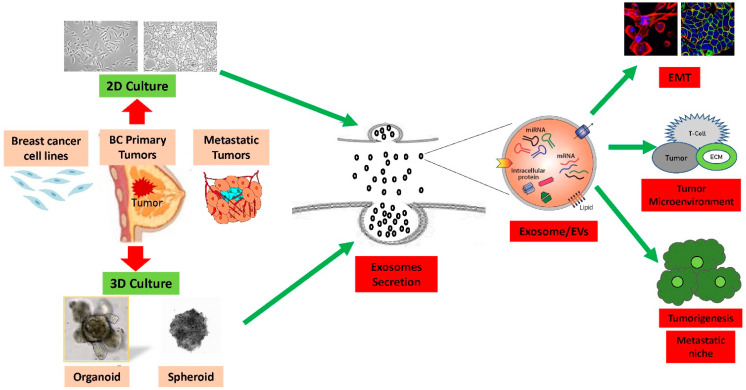
Summary of 2D vs. 3D cultures derived from breast cancer cell lines and tumors: The efficiency of production of exosomes and extracellular vesicle (EVs) are significantly higher in the 3D culture models compared to the 2D culture models. Exosomes contain integrins and other cellular proteins, lipids, nucleic acids, as well as miRNAs, which they transfer into the extracellular microenvironment. Exosomes and other extracellular vesicle cargo induce immune system suppressive pathways due to interaction with extracellular matrix to promote epithelial-to-mesenchymal transition (EMT), breast cancer tumorigenesis, invasion, and metastasis.

**Figure 2 cancers-16-00883-f002:**
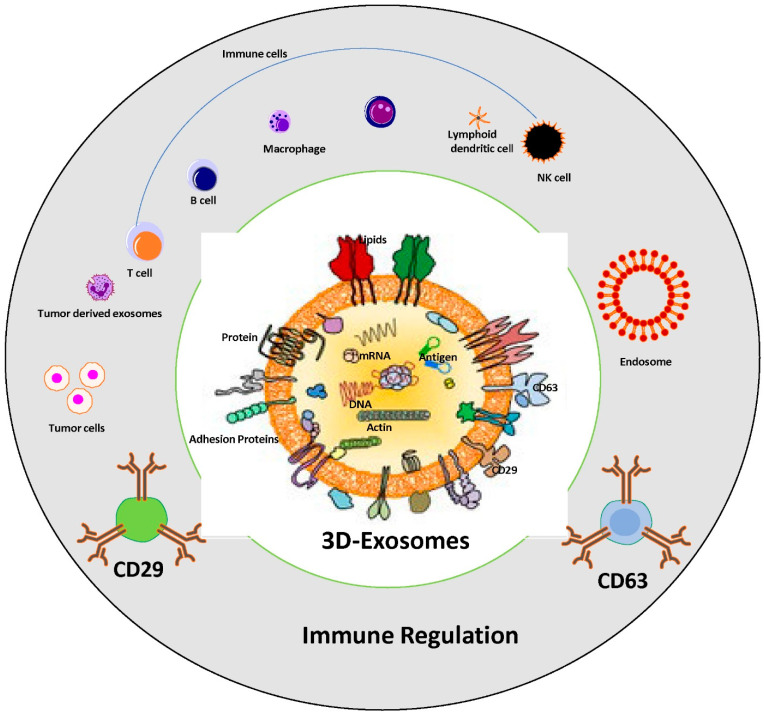
Role of tumor-derived exosomes in immune suppression and cancer progression. Schematic representation of immune regulation system via exosomes; structure of tumor derived exosomes which carry several components, nucleic acids, lipid bilayers and proteins including tetraspanin family, adhesion proteins, antigen binding proteins, etc., are associated with immune cells regulation. The exosomes cargo CD63 and CD29 used as “exosome markers” in cancer progression.

**Table 1 cancers-16-00883-t001:** Exosomes from a 3D culture system of breast cancer and applications.

3D In Vitro System	Types of Cells/Tissues	Characterization Techniques	Functions	Reference
Spheroids	Hs578T cells	Live cell imaging, confocal laser fluorescent microscopy.	To determine TME and the significances of elevated calcium concentrations on spheroids	[16]
Scaffold	MDA-MB-231 and MCF-7	Scanning electron microscopy, immunohistochemistry, immunofluorescence	To determine the oxygen level in tumors and compare the physiologic conditions in microenvironment	[62]
Mammosphere /organoids	MCF10, DCIS	Ultracentrifugation method, fluorescent linker PKH-26 antibody	To find that preadipocyte-derived exosomes promote cancer stem cells and tumorigenesis	[63]
Spheroids	MDA-MB-231	Sequential centrifugation and transmission electron microscopy	To determine EGFR and MET, that promote cancer metastasis via modulation of immune system	[64]
Spheroids	HCC70, HCC1954, and MCF-7	Immunoelectron microscopy	To determine therapeutic effect to target EGFR-nucleic acid drugs	[65]
Spheroids/Organoids	MDA-MB-231, T47DA18, MCF7, HMECs and Primary mammary cells	Electron microscopy, flow cytometry, immunofluorescence microscopy (IFA)	To determine reactive oxygen species which induce DNA damage and autophagy in breast cancer	[66]
Spheroids, Co-culture	MDA-MB-231, MCF7	Transmission electron microscopy	Cancer development and progression via exosomal miR-500a-5p	[67,68]
Spheroids, Organoids	MDA-MB-231, MDA-MB-468, human breast cancer tissues	Ultracentrifugation, transmission electron microscopey, immunofluorescence microscopy	Breast cancer cell glycolysis by sponging miR-1252-5p which regulated PFKFB2 expression	[69]
Spheroids	HCC-1806, HCC 1937, MDA-MB-231, MDA-MB-468	Ultracentrifugation, transmission electron microscopy and flow cytometry	Hypoxia-responsive in TNBC suppression	[70]
Brain organoids on chip system	Treated brain organoids with exosomes extracted from breast cancer cell line MCF-7	Fabrication of micropillar chips, ultracentrifugation, tissue cryosection and immunohistochemistry, transmission electron microscopy	To determine the effects of breast cancer derived exosomes on the neurodevelopment of brain	[71]
Organoids	Murine cells, mouse cell, HEMC	Ultracentrifugation, tissue cryosection and immunohistochemistry, transmission electron microscopy, Odyssey scanner	To determine tumor progression and metastasis by inducing vascular leakiness, angiogenesis, invasion, immunomodulation and chemoresistance	[72]
Organoids, spheroids, and single cell colony assay	Human breast cancer cells from patients, MDA-MB-231, T47D, and MCF7, HTB-26, HTB-133 and HTB-22	Ultracentrifugation, tissue cryosection and immunohistochemistry, transmission electron microscopy	miR-93 functions as a metastasis suppressor by suppressing both invasion ability and CSC properties in breast cancers	[73]
Spheroids	MCF7 breast cancer cells	Ultracentrifugation, transmission electron microscopy, immunocytochemistry	EVs of BC cells collectively supporting cancer cell dormancy	[74]

**Table 2 cancers-16-00883-t002:** Composition of 3D exosomes.

Proteins	Lipid Bilayers	Nucleic Acids
Accessory proteins (Alix, TSG101, HSC70, and HSP90β) [75,76]	Sphingomyelin, cholesterol, ceramide, phosphatidyl choline, andphosphatidylethanolamine [12,76,77,78]	Circular DNA [79],RNAs [80]
Tetraspanin family (D63, CD9, and CD81) [81]		miR-222 [82]
ESCRT proteins and transmembrane [83]		miR-125a-3p [84]
Immunoglobins, intacellular proteins [77]		miR-100 [85]
RNA-binding protein LIN28 [86]		miR-127, miR-7641, and miR-205 [87,88]
Domain proteins, signal transduction, and membrane transport and fusion [89,90,91]

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
