# Peer review of "Advances in 3D Culture Models to Study Exosomes in Triple-Negative Breast Cancer"

_cancers, 2024, doi:10.3390/cancers16050883_

Round 1

Reviewer 1 Report

Comments and Suggestions for Authors

In the submitted mini-review the authors have tried to present the recent advances in 3D culture models to study exosomes in triple negative breast cancer. While the topic is interesting, what I miss most is the methodology section, presenting the process of reviewing the literature, choosing the articles etc. For more details, please look at the PRISMA rules, a direct link can be find in the instructions for authors of works submitted to Cancers.

Other issues are listed below.

Abstract is too long. According to Cancers policy it is limited to 200 words, while right now it is almost twice as much.

In the review there is only one table and solely two figures. This is not enough.

Line 58, it is quite unusual to call the work from 2015 a recent one.

Lines 85-86, it should be highlighted here that only the studies related to triple negative breast cancer has been reviewed for this purpose.

Figure 1, EMT and EVs abbreviations must be explained in the caption to this figure

Lines 146-149, the authors write “recent studies…” and then cite single work from 2003. This should be corrected.

Lines 183-187, the references is missing

Line 187, it should be “O2

Line 187, either “concentration” or “level”

Line 202, this has been already defined in line 97

Line 266-270, why the role of ECM is explained at this point and not earlier, since the ECM has been mentioned a few times above?

Figure 2, this figure is not informative at all. It shows some basic elements of the immune systems without any relations between them. What is the purpose of this figure? And why is “Progression” with capital “P”?

At the end, the part describing each authors individual contribution is missing. This is mandatory in all of the MDPI journals.

Author Response

Response to Reviewer 1

Dear Reviewer,

We are very grateful for the insightful and thorough review of our manuscript.

We have now revised our manuscript by taking in consideration the comments you provided to us. We just would like you to keep in mind that this is a review of the literature that was available to us to discuss the potential roles of exosomes in the pathology of TNBC tumors, as well as the role the 3D culture systems play in affecting the function of exosomes as platforms to drug delivery, response to therapy, and interaction with the tumor immune environment. Because of the nature of this manuscript as a review, we did not see where a methodology section would fit in this manuscript, since no experiments were proposed or conducted.

As for the remaining comments, please find a point-by-point response to each comment.

  • Abstract is too long. According to Cancers policy it is limited to 200 words, while right now it is almost twice as much.

Wee have now revised the Abstract and reduced its size without compromising the message we seek to send the readers. 

  • In the review there is only one table and solely two figures. This is not enough.

We have now added a second table summarizing the molecular composition of exosomes

  • Line 58, it is quite unusual to call the work from 2015 a recent one.

We have now revised this quotation and just referred to “a study that was published in 2015”.

  • Lines 85-86, it should be highlighted here that only the studies related to triple negative breast cancer has been reviewed for this purpose.

We revised this sentence to indicate that the studies were indeed related to TNBC.

  • Figure 1, EMT and EVs abbreviations must be explained in the caption to this figure

We have now expanded the figure legend to spell out EMT and EVs, along with other information shown in Figure 1.

  • Lines 146-149, the authors write “recent studies…” and then cite single work from 2003. This should be corrected.

We have now revised this statement that now reads “another study”

  • Lines 183-187, the references is missing

A reference was added to support the statement.

  • Line 187, it should be “O2”

Corrected

  • Line 187, either “concentration” or “level”

The statement was corrected to emphasize concentration

  • Line 202, this has been already defined in line 97

Corrected

  • Line 266-270, why the role of ECM is explained at this point and not earlier, since the ECM has been mentioned a few times above?

Given the role of the ECM in regulating the tumor microenvironment, we felt that this woold a good place to expand on this function in context of exosomes.

  • Figure 2, this figure is not informative at all. It shows some basic elements of the immune systems without any relations between them. What is the purpose of this figure? And why is “Progression” with capital “P”?

We have now provided a detailed caption to explain the purpose of this figure, namely the potential role of exosomes in the regulation of the several components of the immune system.

  • At the end, the part describing each authors individual contribution is missing. This is mandatory in all of the MDPI journals.

We have now provided this section.

We hope we have addressed all your concerns and that our revised manuscript would now be accepted for publication in Cancers.

With Kind regards

Khalid Sossey-Alaoui

Reviewer 2 Report

Comments and Suggestions for Authors

This manuscript/review article is important for the field of breast cancer research.

Comments:

1. One review article (PMID: 31882647) had some discussion on 3D model in TNBC. What is the difference between your review vs the previous review article? Also please site that article as reference.

2. Please add a Table to summarize the composition of exosomes in 3D model of TNBC.  What would be the impact of those composition of exosomes. also compare the exosomes of TNBC with the exosomes of other types of breast cancers if the data is available.

Author Response

Response to Reviewer 2

Dear Reviewer,

We are very grateful for the insightful and thorough review of our manuscript.

We have now revised our manuscript by taking in consideration the comments you provided to us.

Please find a point-by-point response to each one of the comments you raised.

  1. One review article (PMID: 31882647) had some discussion on 3D model in TNBC. What is the difference between your review vs the previous review article? Also please site that article as reference.

We thank the Reviewer for pointing our attention to this study. We have now cited the impact of this study in our review and included the citation to the reference list (Ref 54).

  1. Please add a Table to summarize the composition of exosomes in 3D model of TNBC.  What would be the impact of those composition of exosomes. also compare the exosomes of TNBC with the exosomes of other types of breast cancers if the data is available.

We have now included a new table where the molecular composition of Exosomes is detailed along with the respective references.

We hope we have addressed all your concerns and that our revised manuscript would now be accepted for publication in Cancers.

With Kind regards

Khalid Sossey-Alaoui

Round 2

Reviewer 1 Report

Comments and Suggestions for Authors

The authors have corrected and revised their work. This version is acceptable.

Reviewer 2 Report

Comments and Suggestions for Authors

No more comments.